# Organophosphorus Flame Retardants: A Global Review of Indoor Contamination and Human Exposure in Europe and Epidemiological Evidence

**DOI:** 10.3390/ijerph17186713

**Published:** 2020-09-15

**Authors:** Zohra Chupeau, Nathalie Bonvallot, Fabien Mercier, Barbara Le Bot, Cecile Chevrier, Philippe Glorennec

**Affiliations:** 1INSERM—Institut National de la Santé et de la Recherche Médicale, F-75000 Paris, France; zohra.chupeau@univ-rennes1.fr (Z.C.); cecile.chevrier@univ-rennes1.fr (C.C.); 2IRSET—Institut de Recherche en Santé, Environnement et Travail, UMR1085, F-35000 Rennes, France; Nathalie.Bonvallot@ehesp.fr (N.B.); Fabien.Mercier@ehesp.fr (F.M.); barbara.lebot@ehesp.fr (B.L.B.); 3EHESP—Ecole des Hautes Etudes en Santé Publique, F-35000 Rennes, France

**Keywords:** environmental health, indoor air quality, epidemiology, biomonitoring, chemical safety, organophosphate ester

## Abstract

We aimed to identify high-priority organophosphorus flame retardants for action and research. We thus critically reviewed literature between 2000 and 2019 investigating organophosphorus flame retardants’ presence indoors and human exposure in Europe, as well as epidemiological evidence of human effects. The most concentrated compounds indoors were tris(2-butoxyethyl)phosphate (TBOEP), tris(1-chloro-2-propyl)phosphate (TCIPP), tris(2,3-dichloropropyl)phosphate (TDCIPP). TBOEP and TCIPP were the most consistently detected compounds in humans’ urine, hair or breast milk as well as tris (butyl) phosphate (TNBP) and tris (phenyl) phosphate (TPHP). Notably, epidemiological evidence concerned reprotoxicity, neurotoxicity, respiratory effects and eczema risk for TDCIPP, eczema increase for TBOEP, and neurodevelopmental outcomes for Isopropylated triarylphosphate isomers (ITPs). Given the ubiquitous presence indoors and the prevalence of exposure, the growing health concern seems justified. TDCIPP and TPHP seem to be of particular concern due to a high prevalence of exposure and epidemiological evidence. TBOEP and TNBP require epidemiological studies regarding outcomes other than respiratory or dermal ones.

## 1. Introduction

Flame retardants are chemical compounds that have, since the 1960s, been added to many products during the manufacturing process [1] with the intention of minimizing the risk of a fire starting, or reducing fire propagation [2]. Polybrominated biphenyls were used in products until they were phased out in 1976. They were replaced by a very similar set of chemicals called polybrominated diphenyl ethers, a family of brominated flame retardants. Polybrominated diphenyl ethers were the most commonly used flame retardants until the early 2000s [3]. They were added to consumer products, including furniture, children’s products, and electronics [4]. Because of their negative impacts on both the environment and health, due to their bioaccumulation and persistence properties, they have been classified as persistent organic pollutants (POPs) under the Stockholm Convention and their use has been restricted in Europe since the 2000s [5,6,7,8]. Organophosphorus Flame Retardants (OPFRs) were used, among their other uses as plasticizers or lubricants, as a replacement for brominated flame retardants to maintain fire safety standards after their phase-out [9,10]. In Europe, the total consumption of flame retardants in 2015 was 498,000 metric tons, of which 18% were phosphorus flame retardants, representing 89,640 metric tons [11]—twice as much as brominated flame retardants. OPFRs are the second-largest flame retardants used in Europe, after aluminum trihydroxide [11]. OPFRs are organic esters of phosphoric acid, containing ether alkyl chains or aryl groups, and may be halogenated or nonhalogenated [4]. Halogenated organophosphates are used as flame retardants, while nonhalogenated organophosphates are mostly used as plasticizers in consumer products, textiles and construction materials [12,13]. More specifically, halogenated organophosphates containing such chlorinated forms as tris(1-chloro-2-propyl)phosphate (TCIPP) or tris(2,3-dichloropropyl)phosphate (TDCIPP) are also widely used in furniture, textiles, building materials, polyurethane foam and electronics. Nonhalogenated flame retardants such as tris(2-butoxyethyl)phosphate (TBOEP), Trimethylphosphate (TMP) or Tris(3,5-xylenyl)phosphate (TXP) are mostly used in floor polishes, coatings, engineering thermoplastics and epoxy resins [10]. All OPFR acronyms used are presented in Appendix A.

OPFRs are not chemically bound in products and may release into the environment via volatilization, leaching and/or abrasion [14]. They are ubiquitous in the environment and, since the early 2000s, can be found in water, biota, sediment and soil; because of their widespread use in consumer products, they are especially present in indoor environments in which they partition air and dust [15]. OPFRs have thus the potential to expose a population via the ingestion of dust, inhalation of air or dermal contact with both.

Toxicological studies have observed hazards including neurotoxic and endocrine effects. TMP and tris (phenyl) phosphate (TPHP) are estrogen receptor agonists [16]. Furthermore, several OPFRs (TPHP, TMP, Ethylhexyldiphényl phosphate (EHDPP), Tri-o-cresylphosphate (ToCP), Isodecyl diphenyl phosphate (IDPP), TDCIPP, and Tris(2-chloroethyl)phosphate (TCEP)) have induced a reduction in the proliferation and growth of human neural stem cells, rat neuronal growth and network activity [17].

Considering current use and human health concerns, with increasing numbers of studies being dedicated to OPFRs, several reviews have already been performed. Although these have focused on just one or two aspects (such as contamination, exposure or health effects), it remains important to consider all aspects in order to characterize the public health concern. As regulations differ from one country or group of countries to another, we focused on Europe with the perspective of identifying compounds of concern for (bio)monitoring or regulation.

We reviewed the literature with the following objectives: (i) to identify the compounds of greatest concern in Europe regarding indoor contamination, human exposure and epidemiological evidence and (ii) to identify priority data gaps in knowledge. We therefore reviewed recent European studies dedicated to indoor contaminations and exposures. Because health hazards are not location-specific, we reviewed the epidemiological evidence without geographic restriction.

## 2. Materials and Methods

The search covered the period from 2000 to 2019. This review has compiled OPFR concentrations of indoor air and dust in Europe, human intake in Europe, human exposure in Europe, and epidemiological studies worldwide regardless of location, given the expected low variability in human response. Because the review is ultimately interested in human health, the search was conducted in the PubMed database using the request: organophosphorus flame retardant*[Title/Abstract] OR OPFR[Title/Abstract] OR phosphorus flame retardant*[Title/Abstract] OR phosphate ester flame retardant*[Title/Abstract] OR organophosphate flame[Title/Abstract] OR phosphate flame retardant*[Title/Abstract] OR organophosphate ester flame retardant*[Title/Abstract] OR organophosphate triester*[Title/Abstract] OR organophosphate ester*[Title/Abstract] OR organic flame retardant*[Title/Abstract] OR PFR*[Title/Abstract] OR OPE*[Title/Abstract] OR phosphate triester flame retardant*[Title/Abstract]. A total of 2037 articles were identified and sorted by a single reviewer. Articles that did not match with the review subject or written in a language other than English were excluded on the basis of the title and abstract. Fifty-two articles were selected and classified according to their subject in five categories: “indoor air concentrations”, “indoor dust concentrations”, “human intake estimations”, “biological measurements” and “epidemiological studies”. With the exception of studies on intake estimations, only studies with at least ten measurements were included, to ensure minimal representativeness.

Regarding publications about indoor air and dust concentration, the following items were retrieved: sampling conditions (country, period and location), sample size, minimal, maximal, mean and median concentrations and references. OPFRs’ median concentrations were compared between studies regardless of collection method. Regarding human intake, this review summarizes and compares the intake (or external doses) estimations of several OPFRs via various pathways (ingestion, dermal absorption and inhalation). Regarding publications on biological measurements, the following items were retrieved: metabolite name, parent compound(s), sampling conditions (country, population, size and year), minimum, maximum, mean and median concentrations and references. Regarding epidemiological studies, the following items were retrieved: country, population, and exposure assessment, compounds of interest, health outcomes observed, covariates and human health findings. Studies reporting only a correlation without assessing adjusted association between risk factors and health outcomes were not included.

## 3. Results and Discussion

### 3.1. Indoor Contamination

Thirty-one studies dealing with the contamination of indoor dust and/or air were identified and reviewed. OPFR concentrations with at least ten measurements were investigated in Belgium, the Czech Republic, Denmark, Germany, Norway, Romania, Spain, the Netherlands and the United Kingdom. We noticed that all studies used samples which were collected after 2006. Five studies collected more than 60 samples. In Denmark, Langer et al. (2016) [18] collected 497 dust samples from homes and 151 from daycare centers. Bergh et al. studied 169 air samples in Swedish homes [19]. Luongo and Östman (2016) [20] collected 62 dust and air samples from homes in Sweden, Fromme et al. (2014) [21] collected 63 air samples from German daycare centers and Xu [22] collected 61 air and dust samples from homes in Norway.

#### 3.1.1. Dust

A total of 20 studies dealt with the contamination of indoor dust [18,19,20,21,23,24,25,26,27,28,29,30,31,32,33,34,35,36,37,38]. They are described in detail in the Appendix A. Twenty-nine OPFRs were measured, and 26 were detected in the indoor dust of daycare centers, cars, private homes, offices and schools. The median dust concentration of compounds ranged from <0.0018 μg/g in homes in the United Kingdom [33] to 1600 μg/g in Swedish daycare centers [19]. Concentrations (for detected compounds) are shown in Figure 1. The ten OPFRs having the highest median concentration, independently of the considered indoor environment, were: TBOEP (median = 1600 μg/g in day care centers in Sweden) > TCIPP (median = 65 μg/g in UK homes) > TDCIPP (median = 31 μg/g in UK cars) > TCEP (median = 30 μg/g in homes in Sweden) > EHDPP (median = 29 μg/g in UK classrooms) > TPHP (median = 9.79 μg/g in homes in the Netherlands) > tris (butyl) phosphate (TNBP) (median = 5.60 μg/g in homes in Sweden) > Tri-iso-butylphosphate (TIBP) (median = 5.3 μg/g in homes in Sweden) > Tricresylphosphate (TMPP) (median = 2.7 μg/g in homes in Sweden) > TPHP (median = 2.58 μg/g in homes in The Netherlands).

The paragraph hereafter focuses with more details on studies published since 2012 and not included in the previous review by Wei [39]. The highest OPFR median concentrations were observed in Germany, the UK, Denmark and Sweden. However, the differences between countries may be related to the sample location (daycare centers, cars, private homes, offices and schools). For instance, the higher sum of OPFR concentrations observed in Sweden was explained by a particularly high median concentration of TBOEP in daycare centers (median = 1600 μg/g [19]). The recent studies in homes were conducted in the UK, Sweden, Norway, the Netherlands, Denmark, Germany, the Czech Republic, Spain and Portugal. Almost the same OPFRs were analyzed in the UK, Sweden, Norway, the Netherlands, Germany and the Czech Republic, although OPFR concentrations were different. In the UK, TCIPP was the most predominant compound, followed by TBOEP, whereas in Norway and Germany TBOEP was the most predominant compound, followed by TCIPP. In Sweden, TBOEP and TCIPP were the two OPFRs having the highest concentration. In the Czech Republic, TCIPP was the predominant but TBOEP was not analyzed. OPFR concentrations were higher in the UK and Sweden than in other countries, with a median concentration of total OPFRs (with different OPFRs between studies) of 49 μg/g in Sweden [20], 79 μg/g [33] and 27.44 μg/g [26] in the UK. TBOEP was the most predominant compound in daycare centers studied in Germany and Denmark, with a median concentration 8.5 times higher in Germany. In Denmark, the TBOEP median concentration was followed by the TCEP, TDCIPP and TCIPP median concentrations, while in Germany other OPFRs made a minor contribution (with TDCIPP not measured). Office samples were collected in the UK and Germany, and TCIPP was the most predominant in both countries. It was followed by TBOEP in Germany (not measured in the UK) and EHDPP in the UK. The EHDPP concentration was ten times higher in the UK than in Germany (median = 1.6 μg/g vs. median = 0.14 μg/g) [26,31]. The sum of the OPFR median concentration in the UK was higher than in Germany, mainly because of the high TCIPP concentration. School samples were collected in elementary all-day Austrian schools—only TCEP was measured, with a median concentration of 2.5 μg/g [32]. Samples were collected from cars in the UK, Germany and Spain. OPFR concentrations in the UK were nine times higher than in Germany, and 15 times higher than in Spain. OPFR profiles between the UK and Germany were similar, with TCIPP making a major contribution, followed by TDCIPP and TPHP [26,31]. Two OPFRs were measured in Spain: TPHP and DPHP. Only TPHP was measured in all three countries.

To conclude on OPFRs in dust, TBOEP, TCIPP and TDCIPP were the most concentrated in Europe. In homes, offices and cars, the predominant compound was TCIPP, whereas in daycare centers it was TBOEP. TDCIPP was found to be much higher in dust from vehicles and offices than in dust from the main living areas. These findings confirm those of previous review by Wei [39], who also mentioned high concentrations of TBOEP in home dust in non-European countries such as Japan. In addition, this update highlights that, in a given environment, OPFR concentrations in dust tend to be higher in the UK than in the rest of Europe, likely in line with more stringent fire safety regulations in the UK.

#### 3.1.2. Air

A total of 10 studies [19,20,21,27,36,37,40,41,42,43] dealt with the contamination of indoor air in Europe and are presented in detail in Appendix A. A total of 16 OPFRs were detected; median concentrations ranged from 0.001 ng/m^3^ in homes in the Czech Republic [36] to 330 ng/m^3^ in offices in Sweden [41]. However, the passive sampling technique (polyurethane foam) used by Vykoukalová may underestimate air concentrations by sampling, principally, the gas phase, contrary to other studies with active sampling. The five OPFRs with the highest median concentrations were: TCIPP (median = 330 ng/m^3^ in offices in Sweden) > TNBP (median = 49 ng/m^3^ in Germany daycare centers) > TDCIPP (median = 28 ng/m^3^ in offices in Sweden) > TCEP (median = 25 ng/m^3^ in daycare centers in Sweden) > TIBP (median = 13 ng/m^3^ in homes in Sweden).

We describe hereafter the most recent (>2012) publications investigating indoor air contamination since the previous review by Wei [39], and have thus not included this in our investigation. In homes, air concentrations were measured in four countries: Sweden, Norway, the Czech Republic and Belgium. The concentrations of detected compounds are displayed in Figure 2. With the exception of Belgium (all median < LOD), countries had similar concentration profiles with a predominance of TCIPP followed by TNBP. However, the median TCIPP concentration was between 2 and 10 times higher in Norway than in Sweden or the Czech Republic. In daycare centers (Figure 2), TNBP and TCIPP were measured only in Germany, with TNBP being 25 times more concentrated than TCIPP. In offices (Figure 2), TCIPP largely predominates in Sweden and Germany, with median concentrations being 10 times higher in Sweden. Samples from schools were collected in Germany and Sweden (Figure 2). In Germany, TCIPP, TNBP and TIBP were measured with a similar contribution of TCIPP and TNBP, while in Sweden, only TPHP was analyzed. No study was found that included air sample measurements from cars.

To conclude on European indoor air, the two predominant compounds in air were TCIPP and TNBP, thus confirming previous findings from Wei, who noticed it was not specific to Europe [39]. TCIPP dominated in homes and offices, followed by TNBP, whereas in German daycare centers TNBP concentrations were higher than those of TCIPP.

Overall, on indoor air and dust contamination, 16 OPFRs were measured in air and 30 in dust. We should note that a comparison between all of these studies might be limited by possible heterogeneity in the detection limits and analytical protocols. We did not consider these elements in detail in this review and assume that among the most recent studies in Europe technical heterogeneities are likely to be small and only slightly influential on the present conclusions. The number of collected samples, the measured compounds and the sampling might also differ between studies. Globally, TBOEP and TDCIPP were more concentrated in dust than in air, in line with their log Koa (13.1 and 10.6, respectively). Conversely, TNBP was present in air with a log Koa of 8.2. TCIPP also has a log Koa of 8.2 and was measured at a high concentration in both indoor matrices. Considering the findings, it would be useful to document more broadly the presence of, at least, these main OPFRs in air and dust of living spaces in Europe, considering more countries, and ideally on representative samples of homes, schools, an daycare centers.

### 3.2. Human Exposure to OPFRs in Europe

#### 3.2.1. Intake Estimation

Because of the frequent occurrence of OPFRs in dust and air, humans are exposed to these pollutants via dust ingestion, dermal contact and inhalation, in addition to dietary intake [39]. Several studies have estimated OPFR intake using a similar concentration in the environment, ingestion rates and time-activity patterns [39].

Dust ingestion has long been considered the most important pathway for OPFR exposure [44,45]. The median ingestion via dust to the sum of OPFRs was estimated at 6.6 ng/kg bw/d for nonworking adults and 128 ng/kg bw/d for children in Belgium [46], at 6.5 ng/kg bw/d for adults and 22.4 ng/kg bw/d for children in Germany [25], at 2.6 ng/kg bw/d for adults and 60.6 ng/kg bw/d for children living in urban areas in Romania [30], and at 8.9 ng/kg bw/d for adults in Norway [37]. Sums of OPFRs inform on global intake; however, comparisons are limited because the included compounds may differ from one study to another. Considering compounds separately, a 2015 study by Brommer and Harrad [26] on the UK population, estimated dust ingestion intake at 0.92 ng/kg bw/d for TCIPP, 0.07 ng/kg bw/d for TDCIPP and 0.03 ng/kg bw/d for TCEP in adults. With a higher frequency of hand-to-mouth behavior in children, Brommer and Harrad (2015) [26] estimated exposure via dust ingestion at 43 ng/kg bw/d for TCIPP, 4 ng/kg bw/d for TDCIPP and 1.7 ng/kg bw/d for TCEP in the UK. Therefore, toddler OPFR intake via dust ingestion was approximately 10 to 20 times higher than in adults in Belgium, Germany, Romania and the UK. Abou-Elwafa Abdallah et al. (2016) [47] estimated dermal absorption in the UK for TCIPP, TDCIPP and TCEP for adults at 3.8, 0.2 and 0.1 ng/kg bw/d and for toddlers at 32.9, 1.6 and 1.5 ng/kg bw/d, respectively. Dermal intakes were thus higher than via dust ingestion for these three OPFRs for UK adults. The contrary was true for UK children, and also for many OPFRs in car dust in Greece [28], notably for TBOEP and TPHP, which are not volatile making the ingestion of dust the main exposure route. Data suggest that inhalation (including both gaseous and particulate phases) exposure may also be an important pathway [48] especially for more volatile compounds such as TCEP or TCIPP [37]. In Norway, the adult median inhalation exposure to OPFRs was 9.3 ng/kg bw/d, which is similar to dust ingestion exposure (8.9 ng/kg bw/d) [37].

Overall, major indoor pathways seem to differ from one OPFR to another. In European populations, dust ingestion was the major exposure pathway for TBOEP and TPHP, while inhalation was the major exposure route for TCIPP and TCEP. Unlike EHDPP, dietary exposure is negligible for TBOEP, TPHP, TCIPP and TCEP [22]. Dermal absorption is a minor pathway for EHDPP, TBOEP, TCIPP, TCEP and TPHP [22]. No European data were found concerning a major pathway for TDCIPP. However, these observations are issued from few studies with diverse protocols, OPFRs of interest and microenvironments; a comprehensive and systematic study on exposure routes of main OPFRs would be of great interest.

#### 3.2.2. Human Biological Measurements of Exposure

A total of 12 studies that reported concentrations of OPFRs or their metabolites (presented in Appendix A with their parent compounds) in more than ten human samples were identified and reviewed [15,21,35,49,50,51,52,53,54,55,56,57]. Metabolites were measured in urine, whereas OPFR parents’ compounds were measured in hair and milk. OPFRs and metabolites analyzed in matrices were not exactly the same between articles. These studies were conducted in Germany, Belgium, Norway, Sweden and Spain. The data are summarized in Figure 3 (only detected chemicals are displayed) and detailed in Appendix A.

A total of 15 metabolites were measured in urinary samples (Figure 3 for detected compounds): Diethyl phosphate (DEP, metabolite of TEP), Bis(2-butoxyethyl) phosphate (BBOEP), Bis(2-butoxyethyl)-(2-hydroxyethyl) phosphate (BBOEHEP) and Di-(2-butoxyethyl) phosphate (DBOEP) (metabolites of TBOEP), Diphenyl phosphate (DPHP, metabolite of TPHP), Diethylhexyl phosphate (DEHP, metabolite of TEHP), Bis-(2-chlorethyl)-phosphate (BCEP) and Di-(2-chloroethyl) phosphate (DCEP) (metabolites of TCEP), Di-n-butyl phosphate (DnBP, metabolite of TNBP), BDCIPP (metabolite of TDCIPP), Bis(1,3-dichloro-2-propyl) phosphate (BCIPP) and Di-(2-chloroisopropyl) phosphate (DCIPP, metabolite of TCIPP), Di-m-cresyl phosphate (DmCP, metabolite of TmCP), Di-o-cresyl phosphate (DoCP, metabolite of ToCP) and Di-p-cresyl phosphate (DpCP, metabolite of TpCP). Samples of urine were collected in Norway, Germany and Belgium from children aged from 20 months to 12 years and adults. Median concentrations ranged from 0.12 ng/L for BDCIPP to 3.2 ng/L for DEP [51,54].

Compounds having median concentrations of metabolites above 1 ng/mL, were TEP > TBOEP > TPHP > TEHP [21,51,54]. In populations including mothers and children (general population, see Figure 3), the metabolite with the highest median concentration was DEP with a median of 3.2 ng/L, followed by DEHP (median = 1.4 ng/L) and DPHP (median = 1.3 ng/L) [54].

In children, the two metabolites with the highest concentration were DBOEP (median = 2 ng/L) [21] and DPHP (median = 1.8 ng/L) [53].

DBOEP, DBP, DPHP and BDCIPP urinary concentrations were measured in samples from both mothers and children [21,50,51,53,57]. DBOEP and DPHP median concentrations were higher in children, whereas DnBP and BDCIPP median concentrations were similar in mothers and children.

The only study to report on OPFR concentrations in hair was conducted in 2012, in Norwegian children aged from 6 to 12 years and their mothers. Seven OPFRs were found: TBOEP, TCEP, TPHP, TDCIPP, EHDPP, TNBP and TMPP (Figure 4). The median concentration ranged from 8 ng/g for TMPP to 318 ng/g for TBOEP [52]. The TBOEP median concentration in children was around five times higher than in mothers, while for other OPFRs the median concentrations were similar. Two studies investigated OPFR concentrations in breastmilk in Sweden and Spain [49,56]. Fifteen OPFRs were measured, ten were detected and their median concentrations ranked as follows: TCIPP > TMP > TBOEP > TNBP > TPHP > EHDPP > TCEP > TDCIPP > TEP > TMPP (Figure 5). TCIPP was predominant, with a median concentration of 45 ng/g of lipid weight. Other OPFR median concentrations ranged from 0.8 to 19 ng/g of lipid weight.

To conclude, OPFRs (or their metabolites) concentrations differ between matrices, likely reflecting different metabolisms. In urine, the OPFR with the highest concentration was DEP (a TEP metabolite), so an estimation of its intake and exposure pathways would be of interest. It is important to stress that DEP is also a degradation product of organophosphorus insecticides such as chlorpyrifos [54]. In children’s hair, the most concentrated OPFR was TBOEP, whose intake was high, especially via dust ingestion. The highest concentration in TCEP was noticed in mothers’ hair, while the highest concentration of TCIPP was in breastmilk. These OPFRs had a high human intake estimation, mainly via inhalation and dust ingestion [22]. Despite the high TDCIPP human intake via dust ingestion and dermal absorption, concentrations of this OPFR (or its metabolite) in urine, hair and breastmilk appeared very low compared to other OPFRs, which deserves confirmation from other studies. Considering their widespread environmental occurrence, the biomonitoring of OPFR is likely to develop in the coming years. A great expectation is a comprehensive strategy on the search for metabolites or parent compounds in human matrices, before representative surveys could occur on large populations.

This broad review allows us to put indoor concentrations and human exposure in Europe into perspective. TCIPP, TDCIPP and TBOEP had the highest median concentrations in dust samples, and logically the highest estimation for dust ingestion intake. Specifically, TBOEP was measured at high concentrations in daycare centers, but only in Germany, and in children’s hair. TCIPP was measured as having a median concentration ten times higher than TBOEP and TDCIPP in maternal milk, consistent with high dust ingestion TCIPP intake.

### 3.3. Epidemiological Associations

Since 2000, 19 epidemiological studies have been assessed, mostly in the USA, on the associations between exposure to OPFRs and health effects in humans, and they are presented in Table 1 [40,58,59,60,61,62,63,64,65,66,67,68,69,70,71,72,73,74]. The health outcomes studied were reproductive, thyroid, neurodevelopmental, respiratory, immunotoxic and dermal. Exposure to OPFRs was assessed using house dust, passive wristband, and urine or blood samples. In these studies, metabolites are specific to the OPFR parent compound, with the exception of the TPHP metabolite, because DPHP is also a metabolite of EHDPP and resorcinol bis(diphenyl phosphate) [58,59].

Reproductive functions: Three OPFR, TDCIPP, TPHP and Isopropylated triarylphosphate isomers (ITPs) (or their metabolites) were tested for possible impacts on the reproductive system. Of 201 couples undergoing in vitro fertilization, the urinary TDCIPP metabolite concentrations among the men (though not the women) were associated with a reduced probability of successful oocyte fertilization [61]. In another study, urinary concentrations of the TDCIPP metabolite were cross-sectionally associated with a decrease in both sperm quality parameters (sperm motility, sperm morphology, straight lie velocity and curvilinear velocity) and the thyroid-stimulating hormone (TSH) levels that prompt the thyroid gland to produce hormones [72]. An inverse association was observed between TDCIPP house dust concentrations and occupants’ prolactin hormonal levels in serum [74]. Among female infants, the maternal TDCIPP and ITP metabolite concentrations measured in urine samples collected during the late-second or early-third trimester were associated with an increased risk of preterm birth [63]. Conversely, among male infants, the maternal ITP metabolite concentrations measured in urine samples collected during the late-second or early-third trimester were associated with a decreased risk of preterm birth [63]. The TPHP metabolite concentrations measured in urine samples collected during in vitro fecundation from women were inversely associated with the probability of successful fertilization (presence of a fertilized oocyte with two pronuclei 17–20 h after insemination), implantation (defined as a serum β-hCGlevel > 6 mIU = mL, approximately 17 d (range = 15–20 d) after egg retrieval), clinical pregnancy (the presence of an intrauterine pregnancy confirmed by ultrasound at approximately 6-week gestation) and live birth (defined as the birth of a neonate on or after 24-week gestation) [61]. Another study observed an association between house dust TPHP concentration and decrease in sperm concentration among 50 men from couples that were infertile due to a male factor, a female factor, or a combination of both [74]. The same study observed an association between house dust TPHP concentration and an increase in serum prolactin hormone levels in a population of 38 men [74]. Among male infants, urinary TPHP metabolite concentrations were associated with a modest increase in gestational duration. Baby boys having the highest levels of prenatal TPHP exposure were born approximately 5 days later than those with the lowest levels of exposure [63].

Thyroid systems: Three OPFRs, TDCIPP, TPHP and TCEP (or their metabolites), were tested for possible impacts on the thyroid system. TDCIPP metabolite urinary concentrations were associated with a decrease in concentrations of total triiodothyronine (T3 hormone that helps regulate various physiological processes including growth, metabolism, body temperature and cardiac rhythm), measured cross-sectionally in serum [72]. A positive association was found between TPHP metabolite urinary concentrations and mean total thyroxine levels measured cross-sectionally in serum (T4 hormone, mainly secreted by the thyroid gland, and essential for proper metabolic functioning) among women, but not in men [67]. The TDCIPP urinary metabolite concentrations measured in women from a case-control study (100 cases and 100 controls) were not associated with an increased risk of papillary thyroid cancer [62]. The TCEP concentrations measured in house dust samples were associated with an increased risk of papillary thyroid cancer in women from a case-control study (70 cases and 70 controls) [65].

Neurodevelopmental outcomes: TDCIPP, TPHP and ITP have been studied to assess their effect on neurodevelopment. Prenatal urinary TDCIPP metabolite concentrations were positively associated with attention problems (measured using the Behavior Assessment System for Children-2 scale—Teacher Report) among 7-year-olds [64]. This study found no association between prenatal urinary TDCIPP metabolite concentrations and the hyperactivity scale among children aged 7 [64]. Prenatal urinary TPHP metabolite concentrations were associated with a decrease in intellectual quotient assessed by the Wechsler Intelligence Scale for Children, 4th edition (WISC-IV), among children at age 7, and particularly for the working memory domain [64]. Prenatal urinary ITP metabolite concentration was inversely associated with age-standardized scores on the MacArthur-Bates Communicative Development Inventories vocabulary assessment, Mullen Scales of Early Learning cognitive composite score, fine motor scale and expressive language scale at age 36 months [58]. Furthermore, prenatal concentrations of this metabolite were positively associated with higher scores on the hyperactivity scale at age 7 [64] in the California area, where Firemaster^®^ 550 (containing ITPs, Great Lakes Solutions, Chemtura Corporation, Philadelphia, PA, USA.) was used.

Respiratory outcomes and immunotoxicity: Three OPFRs, TDCIPP, TCIPP and TNBP, or their metabolites, were tested for possible effects on the respiratory system or immunotoxicity. TDCIPP measured in house dust samples was associated with an increased risk of wheezing in children 7 years of age [59], though not with asthma among children aged 4–8 years [69]. A positive association between urinary TCIPP metabolite concentrations and an increased risk of rhino-conjunctivitis was cross-sectionally observed among children aged 6–12 years [60]. TCIPP measured in house dust two months after birth was not associated with the development of asthma at 4 or 8 years in a case-control study (110 cases and 110 controls) [69]. In the cross-sectional study conducted by Araki et al. [71], the authors did not observe any association between an increased risk of allergic rhinitis in adults and children and TCIPP concentration in house dust. On the contrary, they found a positive association between TNBP concentrations in house dust and risk of asthma among children and adults, as well as an increased risk of allergic rhinitis [60].

Dermal effects: Only one study [60] tested for an association between increased risk of eczema and TDCIPP. Eczema was evaluated using the International Study of Asthma and Allergies in Childhood (ISAAC), while exposure assessed using TDCIPP in house dust and TBOEP urinary metabolites and TCIPP (or its metabolite BCIPP) was cross-sectionally associated with eczema in school-aged children [60].

Other outcomes: Blood concentrations of six OPFRs (TCIPP, TBOEP, TPHP, TEP, TNBP and EHDPP) independently increased sphingomyelin concentrations (which participate in cardiovascular function) in blood among adults [70]. Otherwise, TBOEP indoor concentrations in air were associated with an increased risk of sick building syndrome in Japan [73].

In conclusion, both TDCIPP and TPHP were consistently associated with multiple health outcomes, mainly reproductive. High TCIPP concentrations were possibly associated with adverse respiratory outcomes when exposure was assessed via measurement in human samples, but not when assessed by measurement in indoor environments. Two epidemiological studies associated the ITP urinary metabolite concentrations during pregnancy with various neurodevelopmental outcomes among children in North America. Globally, most epidemiologic studies have addressed relatively modest population sample sizes and thus were limited in statistical power. Considering OPFRs in large epidemiological studies would be useful, especially for compounds widely measured in human environments and matrices such as TBOEP and TNBP.

## 4. Conclusions

Due to an increase in their use, a significant OPFR presence was observed in both indoor environments and human biological matrices in Europe. In this review, we chose to have a broad spectrum from contamination to health effects in humans, complementary to more detailed reviews that exist on specific aspects. We tabulated our findings in Table 2, and summarized them in Figure 6, with the objective to contribute to identifying the prevention and research areas that should be priorities. The most concentrated OPFRs in dust were TBOEP, TCIPP and TDCIPP, and in the air, TCIPP and TNBP. These OPFRs deserve to be studied in more microenvironments in larger and representative surveys. Of these OPFRs, both TBOEP and TCIPP were also found to have the highest concentrations in human matrices, especially among children. TDCIPP and TNBP were also observed in human matrices, however these were among the lowest concentrations. These findings rely on few studies and need to be confirmed on larger representative samples. TDCIPP is one of the most investigated OPFRs in association studies, and has been linked with multiple health outcomes, while TPHP is associated with reproductive outcomes. Considering widespread exposure and epidemiological evidence for these two OPFRs, health risk assessment, including a more systematic assessment of relative importance of different microenvironments and exposure routes, may be useful in informing decision making about preventive actions. While TBOEP and TNBP were frequently found in both indoor environments and human matrices, especially in children, association studies have mostly investigated respiratory outcomes, so epidemiological studies would be a priority for these compounds. Table 2 also indicates that several OPFRs have occasionally been studied and detected indoors, so both confirmation studies and a search in biological matrices would be of interest. Lastly, there is a lack of European data concerning Isopropylphenyl phenyl phosphate (IP-PPP) concentrations in both indoor environments and human matrices that must be investigated. Moreover, its classification as potentially neurotoxic and reprotoxic for the fetus (Vermont Department of Health) might incite further epidemiological studies with enough statistical power to assess a potential effect on neurodevelopmental outcomes. More generally, neurodevelopmental outcomes appear to have been studied less than other health effects and may deserve specific attention.

## Figures and Tables

**Figure 1 ijerph-17-06713-f001:**
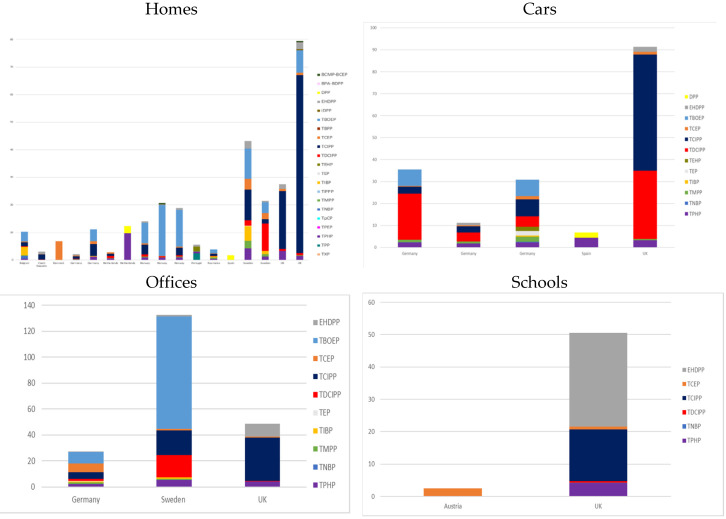
Organophosphorus Flame Retardants (OPFRs) dust median concentration in Europe, 2008–2018 (µg/g).

**Figure 2 ijerph-17-06713-f002:**
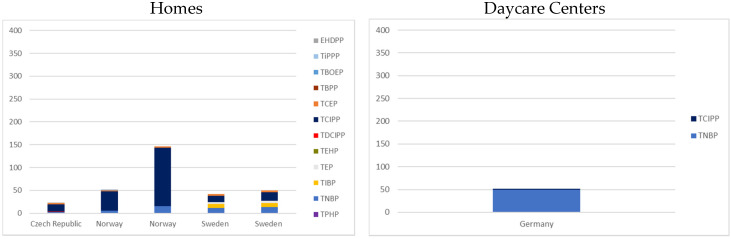
OPFRs air median concentration in Europe, 2000–2018 (ng/m^3^) (one bar per article).

**Figure 3 ijerph-17-06713-f003:**
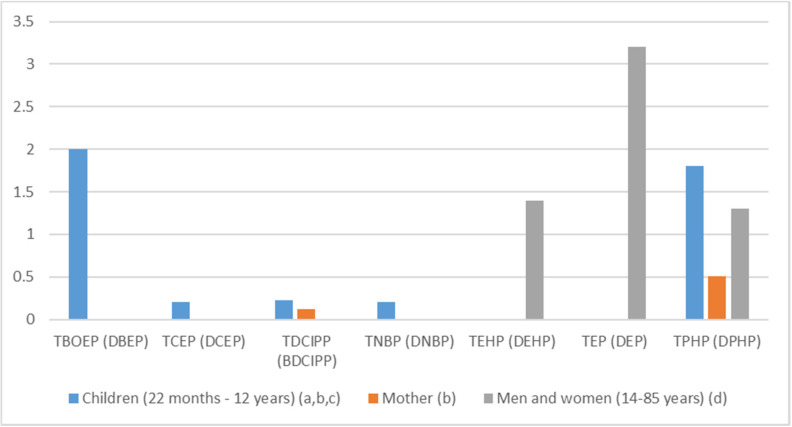
OPFRs (metabolite) highest median urinary concentrations (ng/L) in Europe, 2000–2018 (ng/mL); (**a**) Fromme et al., 2014 [21]; (**b**) Cequier et al., 2015 [27]; (**c**) Larsonn et al., 2014 [53]; (**d**) Reemtsma et al., 2011 [54].

**Figure 4 ijerph-17-06713-f004:**
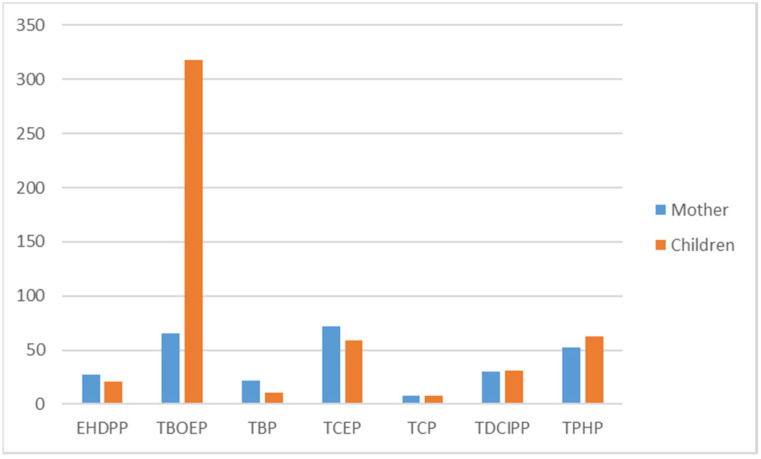
Median OPFR concentrations found in hair in Europe, 2000–2018 [51] (ng/g).

**Figure 5 ijerph-17-06713-f005:**
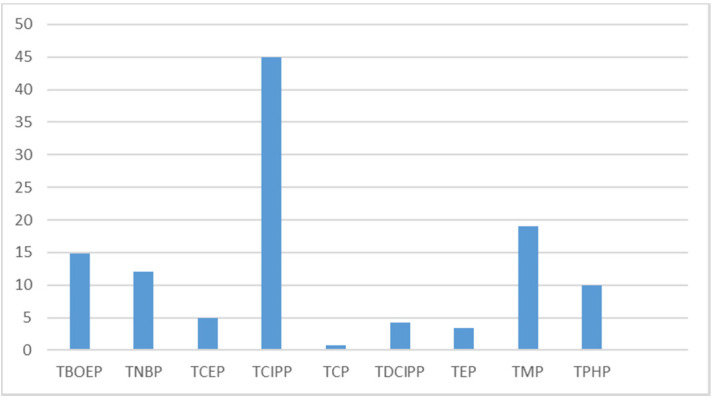
Maximal median OPFR concentrations found in breastmilk in Europe, 2000–2019 (ng/g of lipid weight) [49,56].

**Figure 6 ijerph-17-06713-f006:**
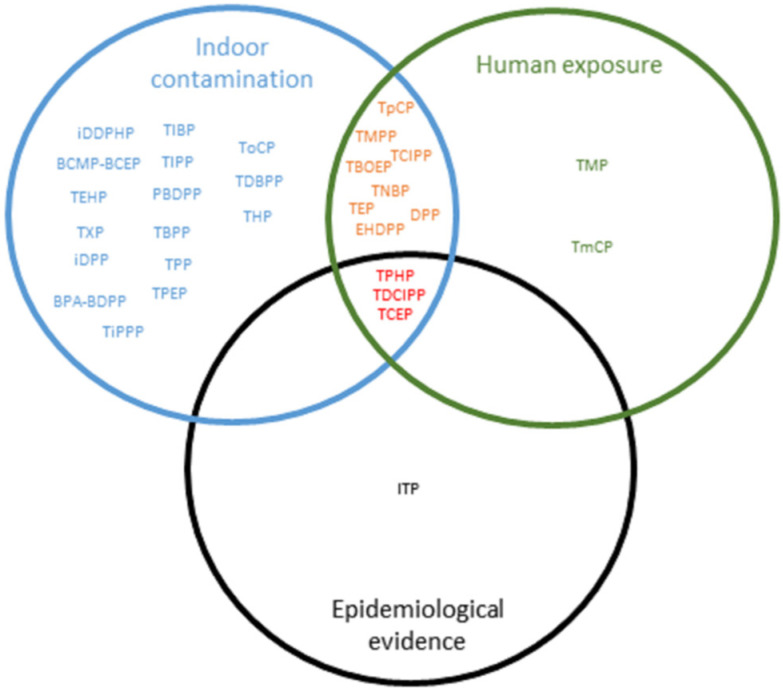
Schematic overview of indoor contamination, human exposure in Europe and epidemiologic evidence of organophosphorus flame retardants. 2000–2019.

**Table 1 ijerph-17-06713-t001:** Epidemiological evidence on organophosphorus flame retardants, 2000–2019.

Author	Date	Country	Population	Exposure Assessment	Compounds of Interest	Health Outcome	Covariates	Human Health Findings
Doherty et al. [58]	2019	USA	149 children of 36 months	Urine sample collected from mothers between 24- and 29-week gestation	DPHP, BDCIPP, IP-PPP, BCIPHIPP	Children’s cognitive function (Composite, Fine Motor, Visual Reception, Receptive Language, Expressive Language) was assessed using the Mullen Scales of Early Learning (MSEL) at age between 2 and 3 years	Maternal age, education, income, race/ethnicity, BMI, and child’s sex	Concentrations of **IP-PPP** (ng/mL) were associated with **MSEL Cognitive Composite Score** (β= −2.61; 95% CI: −5.69, 0.46), **Fine Motor Scale** (β= −3.08; 95% CI: −5.26, −0.91) and the **Expressive Language Scale** (β= −1.21; 95% CI: −2.91, 0.49)
			227 children of 36 months	Urine sample collected from mothers between 24- and 29-week gestation	DPHP, BDCIPP, IP-PPP, and BCIPHIPP	Children’s language (Vocabulary, Grammatical Complexity) was assessed using the MacArthur-Bates Communicative Development Inventories (MB-CDI) at age between 2 and 3 years	Maternal age, education, income, race/ethnicity, BMI, and child’s sex	Prenatal **IP-PPP** concentrations were inversely associated with age-standardized scores on the **MB-CDI Vocabulary** assessment (β= −1.19; 95% CI: −2.53, 0.16)
Ait Bamai et al. [59]	2018	Japan	296 children	House dust samples collected at age 7 of children	TMP, TEP, TPP, TBP, TCIP	Eczema and wheeze were assessed in children aged 7 years using the International Study of Asthma and Allergies in Childhood questionnaire	Sex, household income, maternal smoking, and parental history of atopy.	Among children without any filaggrin mutations, **TDCIPP** was associated with **wheeze** (OR: 1.22, 95% CI: 1.00–1.48)
					TCEP, TEHP, TBEP, TDCPP, TPhP, TCP			
Araki et al. [60]	2018	Japan	128 elementary school-aged children	Multisurface dust	TMP, TEP, TPP, TNBP,	International Study of Asthma and Allergies in Childhood (ISAAC) questionnaire	Sex, grade, annual income, and dampness index	Association between **TDCIPP** in house dust and **eczema** (OR:3.75; 95% CI: 1.39, 10.2)
					TCIPP, TCEP, TEHP, TBEP,			
					TDCPP, TPHP, TMPP			
			113 to 128 elementary school-aged children	Urine samples collected from children	5-HO-EHDPHP, EHPHP,	International Study of Asthma and Allergies in Childhood (ISAAC) questionnaire	Sex, grade, annual income, dampness index, and creatinine	Association between **ΣuTCIPP** and **rhinoconjunctivitis** (4th quartile vs. 1st quartile) (OR= 5.01; 95% CI: 1.53, 6.5; *p* = 0.008], **TBEP-OH** (>LOD vs. <LOD) and **eczema** (OR= 2.86; 95% CI: 1.04, 7.85; *p* = 0.041), **BDCIPP** (3rd tertile vs. 1st tertile) and **at least one of the symptoms** (**wheeze, rhino-conjunctivitis, eczema**) (OR= 3.91; 95% CI: 1.24, 12.3; *p* = 0.019]
					BBOEP, 3-HO-TBEP, BBOEHEP, BCIPP, BCIPHIPP, DPHP, 4-HO-DPHP,			
					3-HO-TPHP, 4-HO-TPHP, BDCIPP, DNBP, uTCEP			
Carignan et al. [61]	2018	USA	201 couples from the Environment and Reproductive Health (EARTH)	One or two spot urine samples per in vitro fertilization cycle	BCIP, BDCIPP, DPHP, IP-PPP, tb-PPP	Proportion of fertilized oocytes, number of best quality embryos, proportion of cycles resulting in implantation, clinical pregnancy and live birth	Year of IVF treatment cycle, primary infertility diagnosis, and maternal urinary PFR metabolites as well as paternal and maternal age, body mass index, and race/ethnicity.	Paternal urinary concentrations of **BDCIPP** were associated with **fertilization** (95% CI: 0.01, 0.12; *p*-trend = 0.06)
		USA	211 women from the Environment and Reproductive Health (EARTH)	One or two urine samples per IVF cycle	BCIP, BDCIPP, DPHP, IP-PPP, tb-PPP	Proportion of fertilized oocytes, number of best quality embryos, proportion of cycles resulting in implantation, clinical pregnancy and live birth	Maternal age, body mass index, race/ethnicity, year of IVF treatment cycle, and primary Society for Assisted Reproductive Technology (SART) infertility diagnosis at study entry	Association between the levels of two individual metabolites (**DPHP and tb-PPP**) and of **total metabolites**, and **reduced probability of successful fertilization, implantation, clinical pregnancy, and live birth**
Deziel et al. [62]	2018	USA	200 women (100 papillary thyroid cancer cases and 100 controls)	Single spot urine samples	BCIPP, BCIHPP, BDCIPP		Age, BMI, education level, family history of thyroid cancer, previous benign thyroid disease, and alcohol consumption	No association between **BCIPHIPP, BCIPP, DPHP, BDCIPP, IP-PPP, tb-PPP** and **papillary thyroid cancer (PTC)**
					IP-PPP, DPHP and tb-PPP			
Hoffman et al. [63]	2018	USA	248 pairs women–child	Urine samples collected between 24–30 weeks gestation	BDCIPP, DPHP, IP-PPP,	Gestational age in days (combination of last menstrual period and earliest-ultrasounds data) and birthweight	Maternal age, race, education, parity, prepregnancy BMI and season of urine sample collection	Among female infants, **IP-PPP** was associated with **birth** (β = −1.00 week; 95% CI: −1.85, −0.15 weeks; *p*= 0.02). Among male infants, **DPHP** was associated with **gestational duration** (β = 0.75 weeks; 95% CI: 0.01, 1.50 weeks; *p* = 0.05)
					BCIPHIPP, BCIPP, tb-PPP			
						Preterm birth (defined as <37 weeks gestation)	Maternal age, race, education, parity, prepregnancy BMI and season of urine sample collection	Among females infants, **preterm birth** was associated with **IP-PPP** (OR: 4.58; 95% CI: 1.23, 17.06) and **BDCIPP** (OR: 3.99; CI: 1.08, 14.78). Among male infants maternal urinary **IP-PPP** concentrations were associated with **preterm birth** (OR: 0.21; 95% CI: 0.06, 0.68).
Castorina et al. [64]	2017	USA	248 to 249 pairs women–child	Urine samples collected during the 2nd prenatal study visit	BDCIPP, DPHP, IP-PPP, tb-PPP	Children’s cognitive abilities was assessed by a single bilingual psychometrician at age 7 using the Wechsler Intelligence Scale for Children, 4th edition (WISC-IV) (Full-Scale IQ, Working memory, Perceptual reasoning, Verbal comprehension, Processing speed)	Maternal education, PPVT scores, CES-D scores, country of birth and prenatal urinary DAP metabolite levels, HOME z-score, language of WISC testing, child sex and age at assessment, and household poverty	Association between **DPHP** and **Full-Scale IQ** (β: −2.9; 95% CI: −6.3, 0.5), **DPHP** and **Working memory** (WISC-IV scale) (β: −3.9; 95% CI: −7.3, −0.5), **ΣPFR metabolites** and **Working memory** (WISC-IV scale) (β: −4.6, 95% CI: −8.9, −0.3).
				Urine samples collected during the 2nd prenatal study visit	BDCIPP, DPHP, IP-PPP, tb-PPP	Children’s behavior was assessed by maternal and teacher report at age 7 using the Behavior Assessment System for Children 2 (BASC-2) (ADHD Index, Inattention DSM-IV, Hyperactive/Impulsive DSM-IV, total subscale DSM-IV) and the Conners’ ADHD/DSM-IV Scales (CADS) (Hyperactivity scale, Attention problems scale)	Sex, age at assessment, maternal country of birth, HOME score at 7-years, prenatal DAPs, and maternal depression and education	Association between **IP-PPP** and **Hyperactivity scale** (BASC-2—Maternal Report (T-score) (β: 2.4; 95% CI: 0.1, 4.7), BDCIPP and **Attention problems scale** (BASC-2—Teacher Report (T-score) (β: 1.1; 95% CI: −0.1,2.3; *p*- < 0.1)
Hoffman et al. [65]	2017	USA	70 cases and 70 controls	Dust samples from homes	TCEP, TCIPP, TDCPP and TPHP	Diagnostic of papillary thyroid cancer (PTC)	Indicator of tumor aggressiveness for FR exposure above the median	Higher levels of **TCEP** associated with increased odds of **PTC** (OR: 2.42; 95% CI: (1.10, 5.33)
Lipscomb et al. [66]	2017	USA	72 children aged 3–5 years	Passive wristband samplers worn continuously for 7 days	TPP, TCIPP, TCEP, TDCPP	Children’s social behaviors were assessed using the Social Skills Improvement System-Rating Scales (SSIS-RS) by their teacher in the preschools they were attending (seven subscales representing positive behaviors: Communication, Cooperation, Assertion, Responsibility, Empathy, Engagement, and Self-Control; four subscales representing behavior problem domains: Externalizing, Bullying, Hyperactivity/Inattention, and Internalizing)	Gender, age, family context, and child’s exposure to adverse experiences	**lnΣOPFR** levels were associated with **responsibility** (β = −0.25, *p* < 0.001) and **externalizing problems** (β = 0.31, *p* < 0.05)
Preston et al. [67]	2017	USA	51 adults	133 urine samples collected at months 1,6 and 12	DPHP	Free thyroxine (fT4), total thyroxine (TT4), total triiodothyronine (TT3), and thyroid stimulating hormone (TSH) in serum samples	Sampling round, time of sample collection, specific gravity-corrected iodine and BDE-47 and sex	**DPHP** was associated with a 0.43 μg/dL (95% CI: 0.47, 1.36) increase in **mean TT4 levels**
			25 women	61 urine samples collected at months 1,6 and 12	DPHP	Free thyroxine (fT4), total thyroxine (TT4), total triiodothyronine (TT3), and thyroid stimulating hormone (TSH) in serum samples	Sampling round, time of sample collection, specific gravity-corrected iodine and BDE-47	**DPHP** was associated with a 0.91 μg/dL (95% CI: 0.47, 1.36) **increase in mean TT4 levels**
			26 men	61 urine samples collected at months 1,6 and 12	DPHP	Free thyroxine (fT4), total thyroxine (TT4), total triiodothyronine (TT3), and thyroid stimulating hormone (TSH) in serum samples	Sampling round, time of sample collection, specific gravity-corrected iodine and BDE-47	No significant association between **DPHP** and **TT4, fT4, TT3, TSH**
Soubry et al. [68]	2017	USA	67 men	Urines samples	BCIP, BDCIPP, DPHP, IP-PPP, tb-PPP	DNA extracted from sperm samples	Age, obesity-status and multiple testing, exposure to monoisopropylphenyl	Association between **BDCIPP, DPHP, IP-PPP and hyper- or hypomethylation** of different genes specific to the metabolites
							diphenyl phosphate	
Canbaz et al. [69]	2016	Sweden	110 children who developed asthma at 4 or at 8 years, matched with 110 controls from a large perspective study	Dust collected from the mother’s mattress two months after childbirth	TCEP, TCIPP, TDCPP, TBEP, TPhP, EHDPHP, mmp-TMPP	Asthma at 4 or 8 years was defined based on at least two of the following three criteria: (i) >1 episode of wheeze in the last 12 months; (ii) a doctor’s diagnosis of asthma; (iii) asthma medicine prescribed occasionally or regularly over the last 12 months		No association between PEFRs concentrations and development of childhood asthma
Zhao et al. [70]	2016	China	154 men and 101 women	One blood sample	TCIPP, TBEP, TPHP, TEP, TNBP, EHDPP	Blood samples	Negative association between EHDPP, TPHP, and TNBP levels and sphingosine 1-phosphate concentration	Association between levels of the six PEFRs and **increased sphingomyelin concentration** (*p* < 0.001 for all OPFRs). The **S1P level in the highest quartile of EHDPP** was **36% lower (95% CI: −39%, −33%; *p* < 0.001)** than that in the **lowest quartile, 16% lower (95% CI: −19%, −14%; *p* < 0.001)** than that in **the highest TPHP quartile**, and **36% lower (95% CI: −38%, −33%; *p* < 0.001)** than that in the **highest TNBP quartile**
Araki et al. [71]	2014	Japan	516 inhabitants (adults and children) in 156 different homes	Floor dust	TMP, TEP, TPP, TNBP, TCIPP, TCEP, TEHP, TBEP, TDCPP, T PHP, TMPP	All inhabitants of each home were asked to complete a self-administered questionnaire participants who reported having received medical treatment for bronchial asthma, atopic dermatitis, allergic rhinitis, and/or allergic conjunctivitis at any time during the preceding 2 years were classified as positive	Gender, age, tobacco smoke, ETS exposure,	Association between **TNBP** in multi-surface dust and **asthma** (OR: 5.34; 95% CI: 1.45, 19.7), **TNBP** in **multi-surface dust and allergic rhinitis** (OR: 2.55; 95% CI: 1.29, 45.01)
							recent renovations, wall-to-wall carpeting, dampness	
							index, hair/fur-bearing pets in the dwelling,	
							mechanical ventilation equipment usage, and total	
							fungi	
Meeker et al. [72]	2013	USA	33 men	Urine samples	BDCIPP, DPHP	Blood and semen samples	Age, BMI, and time of sample collection, abstinence period	Association between **BDCIPP** levels and decreases in **sperm quality** parameters, and concentrations of total **T3** (% change: 6.6; 95% CI: 1.6,12,8, *p* = 0.02) and **TSH** in serum (% change: 40.3; 95% IC: 11.4, 77.1, *p* = 0.006). **DPHP** was
								associated with a 57% (95% CI: −77.8, −18.8, *p* = 0.01) decrease in **sperm concentration** and a 20% (95% CI: –41.1, 0.5) decrease in **sperm motility**
Hutter et al. [32]	2013	Austria	436 children	Air	TCEP, TDCPP	The cognitive evaluation was accomplished by a neurodevelopment test	Social status, gender and region (urban/rural)	Significant correlations of **TCEP** in PM10 and PM2.5 and school dust samples with **cognitive performance. Cognitive performance** decreased with increasing **concentrations of TCEP**
Bergh et al. [40]	2011	Sweden	Adults (men and women)	Air	TEP, TiPrP, TPrP, TiBP, TBP, TCEP, TCIPP, TPeP, THP, TDCPP, TPP, DPEHP, TEHP, TToP, d27-TBP cis			No association between OPFRs levels and reported Sick Building Syndrome symptoms
Kanazawa et al. [73]	2010	Japan	134 adults (70 women and 64 men)	Floor dust	TBP, TBEP, TDCPP		Age (ordinal variable in increments of 10 years), gender, history of allergy, time spent	Association between **TBP** and **mucosal symptoms of Sick Building Syndrome** (OR: 15, 95% CI: 2.7–80), **TBOEP** (OR: 0.3, 95% CI: 0.1–0.7), **TDCIPP** (OR: 2.2, 95% CI: 1.0–4.6)
							at home (h/day; ≤12, >12), and condensation and moldy odor	
Meeker et al. [74]	2009	USA	38 men	House dust	TDCPP, TPP	Serum and semen samples: hormones (Free T4, Total T3, TSH, FSH, LH, Inhibin B, Testosterone, SHBG, FAI, estradiol, Prolactin, Sperm concentration, sperm mobility, sperm morphology)	Age, BMI	Association between **TDCIPP** and **Free T4** (β: −2.8; 95% CI: −4.6, −1.0; *p*: 0.004), **TDCPP** and **prolactin** (β:17.3; 95% CI: 4.1–32.2; *p*: 0.008), **TPP** and **prolactin** (β: 9.7; 95% CI: 2.3,18.9; *p*: 0.02)
			50 men				Age, BMI and abstinence period	Association between **TPP** and **sperm concentration** (β: −18.8; 95% CI: −30.1, −4.5; *p*: 0.01)

Bold: Key elements of the table.

**Table 2 ijerph-17-06713-t002:** Summary of OPFRs indoor contamination, population exposure in Europe and epidemiologic evidence.

Compound	Indoor Contamination	Human Exposure	Epidemiological Evidence of Adverse Effect
Reproductive	Thyroid	Respiratory/Immune	Neuro-Development	Dermal
BCMP-BCEP	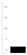						
BDP	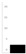				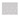		
DPHP	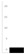	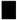					
EHDPP	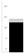						
iDDPHP	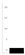						
iDPP	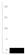						
ITP	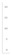		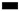			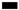	
PBDPP	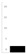						
TBOEP	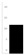	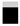			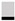		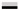
TBPP	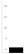						
TCEP	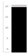	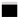			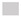	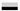	
TCIPP	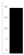	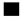	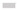	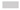	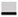		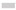
TDBPP	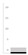						
TDCIPP	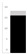	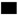	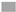	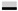	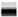	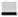	
TEHP	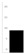				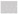		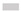
TEP	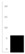	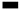			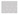		
THP	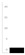						
TIBP	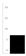						
TIPP	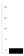						
TiPPP	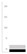						
TmCP	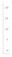						
TMP	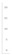				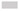		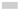
TMPP	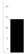	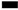					
TNBP	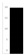	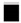			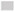		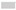
ToCP	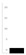	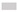					
TpCP	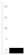	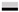					
TPEP	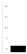						
TPHP	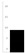	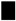	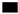				
TPP	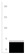		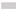				
TXP	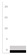

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
