# Peer review of "Organophosphorus Flame Retardants: A Global Review of Indoor Contamination and Human Exposure in Europe and Epidemiological Evidence"

_ijerph, 2020, doi:10.3390/ijerph17186713_

Round 1

Reviewer 1 Report

In this interesting work, the authors reviewed the literature to summarize data about indoor concentration of organophosphorus flame retardants, the exposure of humans to these compounds, and epidemiological evidence on health outcomes. Overall, the review is well written and organized, but a number of inaccuracies can be detected throughout the text.

General comments

Please note that the references format is incorrect. The guidelines are available at: https://www.mdpi.com/journal/ijerph/instructions

Please provide a list of abbreviations.

Specific comments

Line 28. The abbreviation of flame retardants (FRs) should be given on first use (see line 40), and then the authors should aways use the abbreviation (e.g., line 42).

Line 32. Please provide first the full name, then the acronym in parenthesys.

Line 39. Please provide the full name of OPFR. Thereinafter, always use the acronym throughout the text (e.g., line 43).

Line 44. Are the brominated flame retardants mentioned PBDEs?

Lines 47-52. Are non-halogenated compounds also used as flame retardants? In addition, the authors stated first that that “non-halogenated organophosphates are mostly used as plasticizers in consumer products, textiles and construction materials”, then “Halogenated organophosphates… are widely used in furniture, textiles, building materials, polyurethane foam and electronics”, and “ Non halogenated FRs…. are mostly used in floor polishes, coatings, engineering  thermoplastics and epoxy resins”. Please rephrase for more clarity.

Table S2. In English a point, and not a comma, is used for decimal numbers.

Lines 129-130. The value reported in Table S2 is <0,0018. Please amend the text accordingly. In addition, in the table, please correct the reference as Kademoglou et al., 2017a.

Figure 1. The resolution is quite low.

Line 142. Germany or Sweden?

Line 173. Please amend as follows: “These findings confirm THOSE OF previous review by Wei”

Lines 176-177. Do you mean “….in line with more stringent fire safety EUROPEAN  regulations”?

Line 199. In the sense that the studies considered were not included in the previous review?

Line 216. What do the authors mean with: “it was not an European particularity”?

Line 218. In this statement the authors should point out that data are available for only one country.

Lines 237-238. Consider deleting the first “living in urban areas”.

Line 239. Does this measurement refer to adults living in Norway?

Line 246-248. Please specify the country.

Lines 254. Please change “of” to “for”.

Line 260. Why did you write “TBOEP and TPHP”? These compounds were not mentioned in the previous paragraph on dust ingestion.

Lines 266. Table S1 or S4?

Line 272, 281. These statements seem to be in contrast.

In Figure 3 not all metabolites reported in the text (lines 277-283) have been shown.

Lines 282-283. These values seem different from those reported in Table S4 and in Figure 3 (values expressed in ng/ml and not in μg/L, but probably it is a mistake). In addition, the reference Reemtsma et al., 2011 does not find correspondence in Table S4.

In Figure 3 please change BBOEP to DBOEP.

Line 291. In Figure 3, measures of DBOEP, DBP for mothers have not been reported.

Line 294. DnBP is not among the four metabolites cited on line 291.

In the legend of figure 4, please report the unit of measure.

Line 299. The compound TMPP is not shown in Figure 4.

Line 304. TNBP is not reported in Figure 5.

Author Response

Thank you for comments. cf. attached file for answers

Reviewer 2 Report

The authors present a review of research on the environmental exposure to organophosphorus flame retardants in Europe, and of the association of OPFR exposure with human health.  The review is thorough, and the presentation is approachable.  The authors could improve the presentation by addressing several points.

  1. Although the acronym “OPFR” first appears on Page 1, line 39, and is used throughout the paper, it is never defined. I assume it means organophosphorus flame retardant, but the authors should be specific.

  1. The authors present two objectives for the review (page 2, line 74), “i) to identify the compounds of greatest concern in Europe regarding indoor contamination, human exposure and epidemiological evidence and ii) to identify priority data gaps in knowledge.” They do an excellent job of objective 1. However, they do not provide a systematic statement of the gaps in knowledge for either environmental exposure or human health consequences.

  1. The authors note that they searched only the PubMed database. Other relevant databases exist. The authors need to specify not only why they searched PubMed, but also why they did not search these other databases.

  1. The authors note (page 2, line 91) that a single person reviewed all of the 2,037 articles identified. The authors needed to indicate how they checked the validity and reliability of this one individual’s decisions.

  1. The authors need to note if articles in languages other than English were considered for their review, and the justification for any limitations on language

  1. The summary paragraphs at the end of each section of the results make the results understandable. An additional paragraph at the end of each section on knowledge gaps would address the second objective.

  1. The Conclusion section is short and does not include a summary of knowledge gaps.

Author Response

(The authors gave the same response as above.)

Round 2

Reviewer 1 Report

The authors addressed my previous concerns and the revised version of the manuscript is suitable for publication. 

Reviewer 2 Report

The authors have responded appropriately to all of my concerns.